# Colorectal Cancer—The Worst Enemy Is the One We Do Not Know

**DOI:** 10.3390/ijerph20031866

**Published:** 2023-01-19

**Authors:** Marzena Furtak-Niczyporuk, Witold Zardzewiały, Dawid Balicki, Radosław Bernacki, Gabriela Jaworska, Marta Kozłowska, Bartłomiej Drop

**Affiliations:** 1Department of Public Health, Medical University of Lublin, Chodźki Street 1, 20-093 Lublin, Poland; 2Student Scientific Association, Department of Public Health, Medical University of Lublin, Chodźki Street 1, 20-093 Lublin, Poland; 3Department of Medical Informatics and Statistics with E-Learning Laboratory, Medical University of Lublin, 20-059 Lublin, Poland

**Keywords:** colorectal cancer, questionnaire, knowledge, awareness, screening

## Abstract

Background: Colorectal cancer is one of the most common cancers in humans. It is the third most frequently diagnosed malignant neoplasm and is the second highest cause of cancer mortality in the world. Every year, more and more people die of colorectal cancer because the diagnosis is conducted too late. This shows how important a role screening tests and the awareness of the population about the symptoms play in this aspect. This article aimed to determine the knowledge of the Polish population about morbidity, symptoms, prevention, and subjective feelings about the level of availability of knowledge about colorectal cancer. Methods: In 2020, a study was conducted using an online questionnaire assessing the awareness of the Polish population about colorectal cancer. A self-authored questionnaire including questions about socio-demographic characteristics, and 18 questions related to substantive issues, was used. A research group was selected (*n* = 633). The substantive part of the questionnaire included questions examining the respondents’ knowledge about morbidity, symptoms, prevention, and subjective feelings about the level of availability of knowledge about colorectal cancer. Results: The respondents’ awareness level was influenced by demographic factors, such as gender: (*p* < 0.05) and age (*p* < 0.05) and social factors, such as: level of education (*p* < 0.05) or professional situation (*p* < 0.05). Compared to thematic articles from other countries, the research group was divided into smaller subgroups due to the abovementioned factors, due to which it was possible to stratify and analyze the significance of differences between them.

## 1. Introduction

Colorectal cancer (CRC) is the third most frequently diagnosed cancer among men and the second most frequently diagnosed cancer among women. In 2020, there were almost 1.9 million cases worldwide, and it is one of the most common causes of death among cancer patients in the world. CRC is a cancer whose incidence is systematically increasing throughout the year. The risk of developing this disease varies around the world and it differs depending on the geographic region; however, it is more common in middle and highly developed countries [1]. The main cause of high mortality among patients is the lack of clear symptoms that could indicate the presence of this disease. The incidence increases with age, and the likelihood of developing CRC is higher in people with a family history of the disease [2].

From 1990 to 2019, the global age-standardized incidence rate for colorectal cancer increased from 22.2/100,000 to 26.7/100,000, while the global age-standardized mortality rate for CRC decreased from 14.3/100,000 to 13.7/100,000. In 2019, East Asia was the most affected region, with 637,096 new cases and 275,604 deaths. In the same year, Australasia presented the highest age-standardized incidence rate (48.3/100,000), while Central Europe had the highest age-standardized mortality rate (23.6/100,000). The lowest age-standardized incidence rate was in Central Sub-Saharan Africa (7.7/100,000) and South Asia (8.3/100,000). South Asia also had the lowest age-standardized mortality rate (7.3/100,000) [3]. In addition, between 2005 and 2018, the incidence of CRC in Brazil more than tripled from 14.6 to 51.4 per 100,000 population, especially among those aged 50–69 [4]. In addition, an increase in the incidence of colorectal cancer in people younger than 50 has been observed over the past 20 years in the United States [5] and Canada [4].

Compared to other European countries, Poland has the highest CRC incidence rate. Mortality among patients is approximately 10,500 people/year [6]. Colorectal cancer currently ranks first among malignant neoplasms diagnosed in Poland. Among the diagnosed cases, it ranks third among men and second among women in Poland [7].

Reviewing the previous studies about knowledge of the Polish population about colorectal cancer, there were few similar studies. One of them focused on young adults and another on patients in the Department of Gastroenterology and General Oncology, Medical University of Lodz for gastrointestinal diseases. Both pieces of research were based on a self-authored questionnaire. Our study is wider, focusing on a larger group of people of different ages, with statistical elaboration and defines a correlation between the social and professional factors.

### Screening

The primary test for secondary CRC prophylaxis is a colonoscopy [8,9]. This examination involves viewing the large intestine from the lumen side using a thin, flexible tube (colonoscope) with a camera on the end, through which the physician obtains an image of the intestinal mucosa on the monitor. The endoscope is inserted through the rectum and moved to the mouth of the small intestine [10].

A colonoscopy, for most patients, is a painless examination; a small number of people may experience discomfort in the form of distension, flatulence or pressure on the abdomen or, extremely rarely, perforation of the colon, and even death [11].

Since 2000, the CRC Screening Program has been introduced in Poland as part of the National Program for Combating Cancer Diseases, which includes free prophylactic colonoscopies [12]. Its goal is to detect cancer at an early stage of development, increase the cure rate, reduce the incidence and mortality of colorectal cancer. In the endoscopy examination, people aged 50–65, regardless of family history; 40–49 years of age, with a family history of colorectal cancer in a first-degree relative; 25–49 years of age, in a family with Lynch syndrome; and 20–49 years of age, in a family with familial adenomatous polyposis syndrome (FAP), are included [13]. The recommendations of the National Cancer Registry indicate the need for a screening colonoscopy to reduce the risk of incidence and mortality due to CRC by more than 50% [6].

The aim of this study was to examine the awareness of the Polish population about the risk factors and symptoms of colorectal cancer, and to examine the possible correlations between awareness and demographic and social factors. The available statistics mercilessly show the severe scale of the CRC incidence problem, which affects not only the Polish population, but also the societies of other developed countries. These pessimistic data should motivate governments in developed countries to undertake extensive programs to raise citizens’ awareness regarding alarm symptoms, diagnostic options, and genetic predisposition of CRC.

## 2. Materials and Methods

### 2.1. Participants

The statistical analysis included all 633 correctly completed questionnaires out of 1389 (45.6% completed questionnaires) people who initially received questionnaires to complete. Assuming that, according to the main statistical office in Poland, the population of Poland in 2020 was 38,268,000 people, and with the confidence of the public at the level of 98%, and the maximum statistical error at the level of 5%, the minimum number of answers recorded in the questionnaire of the study is 541.

The research group consisted of 414 women and 219 men selected by the simple random sample method.

### 2.2. Study Design

The study, on the basis of a questionnaire assessing the awareness of the Polish population about colorectal cancer, was conducted from 16 March 2020 to 1 May 2020. A free Google disk form was used to collect information. The research tool was a self-authored questionnaire. The questionnaire included the participants’ answers in terms of their gender, age, place of residence, education, and professional situation. The substantive part of the questionnaire included 18 questions examining the respondents’ knowledge about morbidity, symptoms, prevention, and subjective feelings about the level of availability of knowledge about colorectal cancer in Poland. Twelve questions in the knowledge part were single-choice closed questions, two of them were filtering questions, and four of them were closed multiple-choice questions (no. 12, 13, 16, 19), while three of them were stratified by point. For each correct answer, the respondent received 2 points, and for each incorrect answer, the respondent lost 1 point. The appropriate score was assigned to the indicated range of points described in words, determining the respondent’s knowledge on a given topic (from very bad to very good). In order to verify the previously formulated hypotheses, the post hoc Bonferroni test (3.3, 3.4), the Kruskal–Wallis (3.3, 3.4) test, and the chi-square (3.5, 3.6, 3.7) test were performed.

The survey was completely anonymous and voluntary. Before the survey, the respondents were informed about the purpose of the survey.

## 3. Results

The characteristics of the studied group in terms of sociodemographic features are presented in Table 1. 

Approximately 65% of the respondents were women. The age of almost half of the participants of the study was between 20 and 35 years of age. The second largest group in terms of age was represented by the range of 36–50 years (25.1%), then 51–65 years (17.1%), then over 65 (5.4%), and under 20 years of age (3, 2%). More than half of the respondents (50.7%) described their education as college-level. The second most frequent group in terms of education were respondents with secondary education (42.7%), followed by vocational education (5.1%). Only almost two percent (1.6%) of the respondents described their education as primary. A total of 40.1% of the respondents were pupils/students, 34.3% of the respondents were white-collar workers, 16.1% were blue-collar workers, 7% were retirees and pensioners, and 2.5% were unemployed. Almost 70% of the respondents lived in the city, 17.4% in a city with less than 50,000 inhabitants, 13.4% inhabitants in a city between 50 and 100,000 inhabitants, and 38.9% inhabitants in a city over 100,000 inhabitants.

### 3.1. The Sources of Respondensts’ Knowledge of the Colorectal Cancer

The most frequently used source of information about colorectal cancer was the Internet (66.10%). In second place, the respondents used medical books (53.77%), and in third place, they drew their knowledge from medical staff (48.63%). Less frequent choices of the respondents included family or friends (22.60%), press (20.89%), and television or radio (19.86%) in seeking information about colorectal cancer, as shown in Figure 1.

### 3.2. Respondents’ Knowledge about Colorectal Cancer Screening Tests

According to the respondents, the most common examination used in colorectal cancer is colonoscopy (94.53%). Abdominal ultrasound was selected as the second most common examination (46.56%). The least frequent were computed tomography (23.08%), magnetic resonance imaging (18.42%), and gastroscopy (16.60%), as shown in Figure 2.

### 3.3. Colorectal Cancer and Its Correlation with Patient Age according to Respondents’ Knowledge

The vast majority of the respondents (95.90%) saw a correlation between the increased risk of developing colorectal cancer with age, while only 4.10% did not, as shown in Figure 3.

### 3.4. Age, Gender, and Education Significantly Differentiate Knowledge about Colon Cancer Screening

Significant statistical differences were noted (*p* < 0.05) between age, gender, and education as social factors that influenced the results obtained on colorectal cancer screening.

Comparing (36 comparisons; *p* < 0.014) the relevant subgroups against each other yielded surprising results. Men between the ages of 51 and 65 with a college education showed greater knowledge in a statistically significant manner than both younger men with the same level of education, less education, and women.

### 3.5. The Work Situation Significantly Differentiates the Knowledge about Symptoms

There is statistically significant evidence (*p* < 0.05) to suggest that the occupational situation of the respondents influences the knowledge of the alarm symptoms of colorectal cancer.

Significant statistical differences were noted between subgroups. White-collar workers, and pupils and students were more correct (*p* < 0.05) in indicating alarm symptoms, in contrast to the unemployed, pensioners, and blue-collar workers. In terms of knowledge, people with different situations differed in a way statistically (*p* < 0.05). The vast majority (87.20%) correctly consider blood in the stool to be a symptom of colorectal cancer. The second most numerous symptom (61.45%) was weight loss without lifestyle changes. In turn, 58.14% claim that abdominal pain is a symptom of colorectal cancer. Slightly more than half (51.82%) believe that the symptoms include “pencil” stools. A definite minority marked crotch pain (9.79%) and urinary incontinence (3.79%) as symptoms of colorectal cancer, as shown in Figure 4. 

The largest percentage of respondents (34.00%) showed good knowledge about colorectal cancer. Slightly fewer respondents (24.80%) showed average knowledge, and 21.50% showed very good knowledge. Unfortunately, 19.60% of the respondents showed poor knowledge, and only 0.20% showed very poor knowledge about colorectal cancer, as shown in Figure 5.

### 3.6. The Professional Situation Significantly Differentiates the Knowledge about the Occurrence of Bowel Cancer

A significant relationship (*p* < 0.05) was documented between occupational situation and knowledge of colorectal cancer incidence by gender. Pupils and students among the specified subgroups showed the highest knowledge by indicating the correct answer most often. It is likely that educational activity had a significant effect on knowledge of cancer incidence. The worst performers were pension recipients indicating the correct answer only at the level of 29.5%, as shown in Figure 6.

### 3.7. Gender and Education Significantly Differentiate the Knowledge of Cancer Occurrence

Among men with higher education, the opinion about genetic factors predisposing to the occurrence of bowel cancer was recorded as 83.9%, among women with higher education as 93.4%, among men with secondary education as 89.7%, and among women with secondary education as 96.5%. The obtained test result (*p* < 0.05) is statistically significant.

There is a significant relationship (*p* < 0.05) between gender and education and the knowledge of genetic factors predisposing to the occurrence of bowel cancer. Women were generally more likely to indicate the correct answer than men. Similarly, women, both with higher and secondary education, indicated the correct answer with a similar frequency. Women, irrespective of their education, more often had a greater knowledge of genetic factors than men, irrespective of their education.

### 3.8. Age Significantly Differentiates the Knowledge about the Most Common Location of Cancer

We found that 40% (*n* = 8) of people aged <20 correctly identified the most common location of colorectal cancer, which is the rectum (52.1% of all colorectal cancers), as shown in Figure 7. 

Subsequent age groups successively fared better relative to each other: 17.9% (*n* = 312), 20.1% (*n* = 159), 35.2% (*n* = 108), and 55.9% (*n* = 34).

Those in the 65+ age subgroup showed the greatest knowledge of the most common location of colorectal cancer, i.e., the rectum. Due to the fact that those aged 20–35 largely indicated correct answers, no age-related upward trend could be identified.

## 4. Discussion

The main sources from which the respondents obtained information were, respectively: the Internet (66%), books on medical topics (53.4%), and medical personnel (48.3%). Family/friends (22.4%), work (21.1%), and television/radio (19.7%) were used much less frequently as sources of knowledge about CRC. The results obtained in the study conducted on the Hungarian population aged 40–70, analyzing the awareness of the Hungarian population about CRC, showed a completely different distribution of the frequency of using information sources about colorectal cancer. General practitioners or specialists (36.2%) were ranked first, followed by television (35%), followed by the press and brochures (24.6%), followed by the Internet, friends or colleagues, other health care workers, and family. The surprising information found in the study by Gede et al. is that 13.3% of respondents had never heard of colorectal cancer before [14].

The respondents were asked to indicate screening tests that are performed as part of the colorectal cancer prophylactic tests. In the question, the respondents had the option of selecting more than one answer. The most frequently used examination was colonoscopy (94.4%). This may be related to the Colon Cancer Screening program conducted in Poland since 2000. The goal of the program is to reduce the number of cases and deaths from colorectal cancer by performing free prophylactic colonoscopies [12]. Only 22.6% of the respondents knew how long it took to perform another colonoscopy if the previous test result was normal. Moreover, the study shows that men with higher education were characterized by a greater level of knowledge about CRC screening tests, both than men with lower or the same education, and than women. Compared to our study, a population-based study in Saudi Arabia of more than 5000 people found no difference in the knowledge between female and male colon cancer screening [15].

In a cross-sectional study of 1150 people in the Hungarian population, colonoscopy was also the most frequently chosen answer concerning the CRC screening question. There was a low awareness rate in the study regarding the recommended frequency of CRC screening (22.4%). In total, 27% of respondents had never heard of colon cancer screening before, most of them were relatively young and male, and had a low level of education [12].

Out of the 633 responses, 95.8% believed that the risk of developing colorectal cancer increased with age. For comparison, in a study of 308 patients at a medical clinic at Serdang Hospital in Malaysia between 1 April and 31 September 2016, 55.5% of respondents stated that the incidence of colorectal cancer did not correlate with age [16].

A study conducted in 2016 on 200 members of the Polish population showed that its respondents did not fully notice the relationship between gender and the incidence of this type of cancer. Some respondents—26.5% (53 people)—believe that women more often suffer from colorectal cancer, and yet the probability of developing the disease in men is about 1.5–2 times higher and increases with age [17]. For comparison, in our study conducted on 633 people, as many as 53.1% of respondents (338 people) stated a higher incidence of CRC in men, while 8.2% (52 people) believed that this cancer affects women more often.

In a study analyzing the knowledge and awareness of colorectal cancer in the Hungarian population, 69.6% of respondents knew how important the early detection of CRC is, in contrast to our study, in which 95.1% of respondents believed that early detection of colorectal cancer is important and enables its complete cure [14]. There is also a significant discrepancy in the risk of inheriting the predisposition to developing CRC. In a study conducted on the Hungarian population, only 53.0% of people with a positive family history of CRC considered heredity as a risk factor, while in our study, as many as 90.6% of respondents claimed that genetic factors predispose to colorectal cancer [14].

The respondents had a choice of symptoms related to colorectal cancer. In our study, 552 people (86.8%) indicated blood in their stools as the main symptom of CRC, and 389 people (61.2%) chose weight loss without lifestyle changes as the second most frequent symptom. Similarly, in a study in the Kingdom of Bahrain, the symptom of blood in the stool and weight loss was indicated by less than a quarter of local respondents [2].

On the other hand, several studies reported a statistically significant relationship between the respondents’ level of education and awareness of CRC symptoms. A study of 431 respondents in Brunei Darussalam showed a very low level of knowledge of residents about colorectal cancer, as more than half of the people were unable to name any of the symptoms. Despite the astonishing results, non-Malay ethnic groups had higher scores on symptoms of the condition. The results improved as the level of education of the respondents increased [16]. Another study illustrating the differences in responses depending on education is the study of the Jordanian population involving 600 participants. The results clearly showed that better educated people are characterized by greater knowledge about the symptoms of CRC [18]. Another study of 10,078 respondents in Hong Kong showed significantly higher symptom knowledge scores for higher educated and higher earning groups than those with primary education [19]. In the next study, 1019 people in Great Britain were surveyed, where significant differences were also noted. Young people <25 years of age and white-collar workers had a significantly better understanding of CRC symptoms than manual workers [20]. Similarly, in our study, there were also significant static differences between white-collar workers and students who obtained higher scores, including those who showed greater knowledge about the symptoms of CRC, than other groups (blue-collar workers, the unemployed, and retirees and pensioners).

Another study in Saudi Arabia showed that only 15.2% of respondents had already been screened for colorectal cancer, with colonoscopy being the most used method (72.7%) [11]. For comparison, in our study, 152 respondents (29.3%) had experienced a colonoscopy, which is almost twice as many.

## 5. Conclusions

On the basis of the hypotheses and the analysis of statistical data, we obtained appropriate conclusions. They show that age, gender, and education have a significant impact on the knowledge of screening in colorectal cancer. Significantly higher results were achieved by older men aged 51–65 with higher education than younger men with the same or lower education. Men between 51 and 65 years of age have also been found to be more knowledgeable about colon cancer screening than women of the same age. Knowledge about the symptoms and prevalence of colorectal cancer depended on their work situation. The greatest knowledge about the symptoms was shown by white-collar workers and pupils/students. On the other hand, the best results concerning the incidence of colorectal cancer in women and men were obtained by school and university students. The worst, both in terms of the knowledge of symptoms and the occurrence, were retirees and pensioners, who are the economically inactive group. The knowledge about the most common locations of colorectal cancer depending on the age of the examined person is ambiguous. It is not possible to demonstrate an age-related growth relationship, as people under 20 years of age received a significantly high percentage of correct answers. However, people from subsequent age groups achieved higher and higher results with increasing age, and the highest among all examined age groups were obtained by people >65 years of age. It can therefore be concluded that such a high result in people <20 years of age results from the current emphasis on education and prevention of colorectal cancer. Therefore, it is necessary to organize large-scale educational programs for people in their 20s who show ignorance about colorectal cancer prevention, creating posters and information leaflets to educate the public. Foundation nonprofits and the government could be important elements in promoting this knowledge. In response to the question about the correlation between the occurrence of colorectal cancer and predisposing genetic factors, women coped better. The question analyzed the knowledge of this subject depending on gender and education. Both women with higher and secondary education achieved higher statistically significant results than men with the same level of education. The limitation of our survey was the lack of person verification and the inability to limit the influence of external factors on respondents’ answers.

## Figures and Tables

**Figure 1 ijerph-20-01866-f001:**
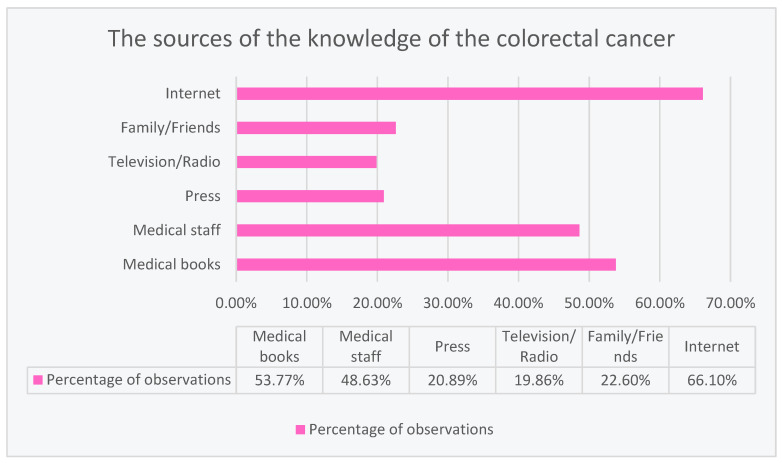
The sources of respondents’ knowledge of the colorectal cancer.

**Figure 2 ijerph-20-01866-f002:**
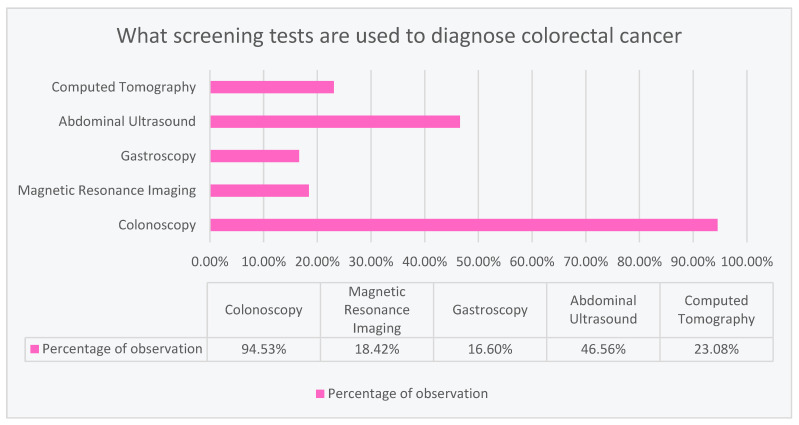
Colorectal cancer screening tests according to respondents’ knowledge.

**Figure 3 ijerph-20-01866-f003:**
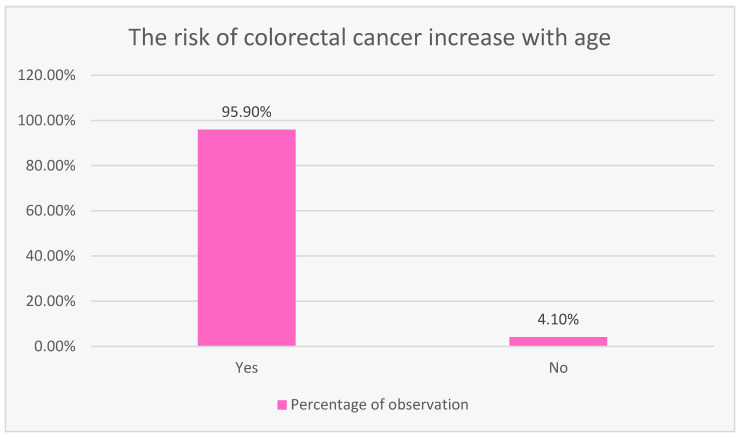
Colorectal cancer and its correlation with patient age according to respondents’ knowledge.

**Figure 4 ijerph-20-01866-f004:**
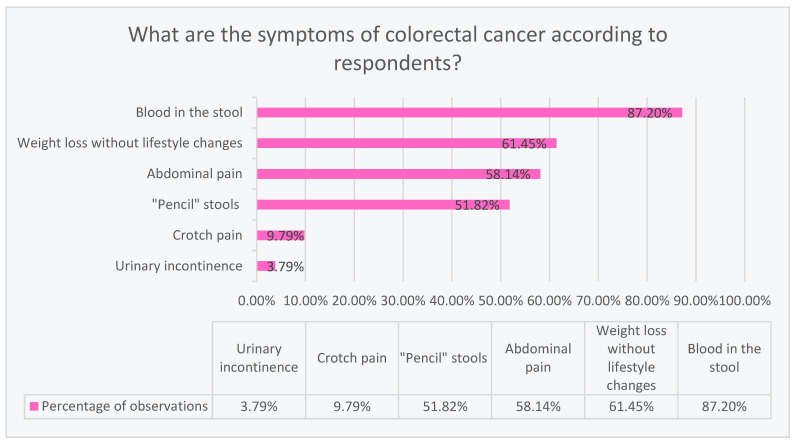
Symptoms of colorectal cancer according to respondents’ knowledge.

**Figure 5 ijerph-20-01866-f005:**
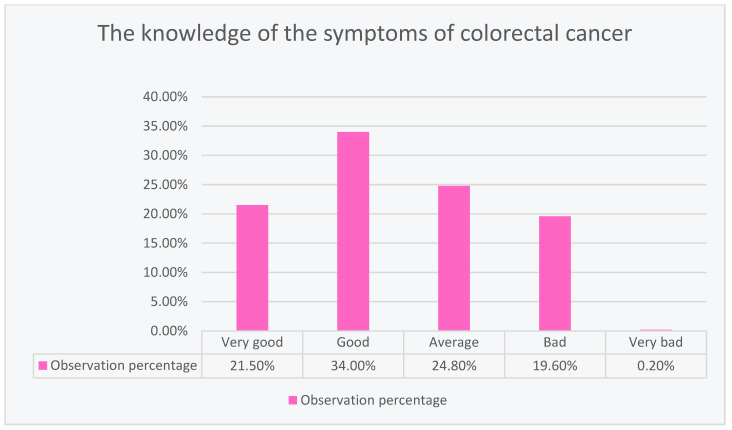
Descriptive scale of respondents’ knowledge about colorectal cancer symptoms.

**Figure 6 ijerph-20-01866-f006:**
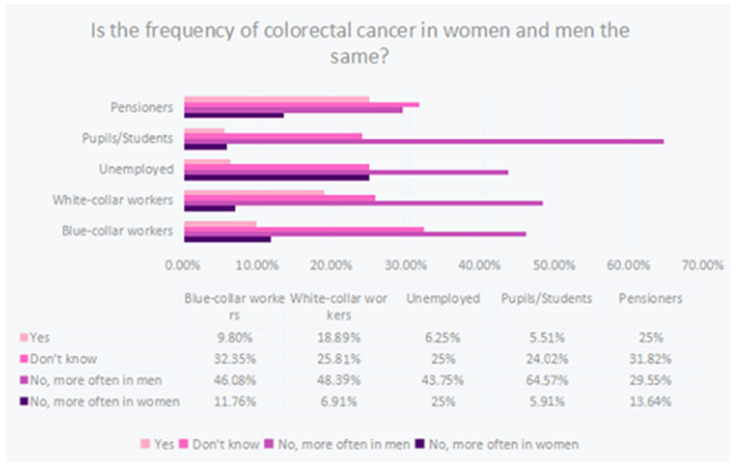
Frequency of colorectal cancer in women and men according to respondents’ knowledge (specifying professional groups).

**Figure 7 ijerph-20-01866-f007:**
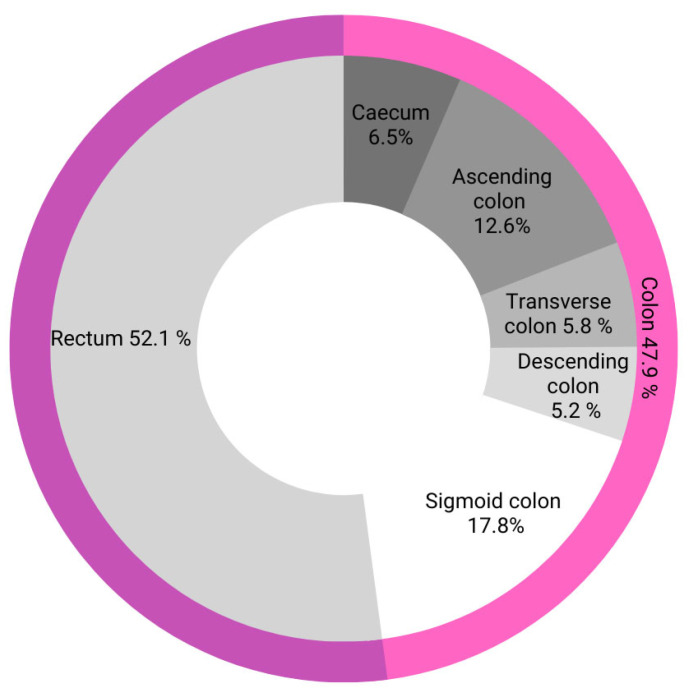
Localization of colorectal cancer with specific anatomical consideration.

**Table 1 ijerph-20-01866-t001:** Characteristics of the studied group.

Sex	Prevalence	Percentage
Male	219	34.6%
Female	414	65.4%
Age	Prevalence	Percentage
Under 20	20	3.2%
20–35	312	49.3%
36–50	159	25.1%
51–65	108	17.1%
Under 65	34	5.4%
Education	Prevalence	Percentage
Primary	10	1.6%
Professional	32	5.1%
Secondary	270	42.7%
College	321	50.7%
Employment situation	Prevalence	Percentage
Blue–collar worker	102	16.1%
White–collar worker	217	34.3%
Unemployed	16	2.5%
Student	254	40.1%
Pensioner	44	7.0%
Residence	Prevalence	Percentage
Village	192	30.3%
City with <50 thousands residents	110	17.4%
City with 50–100 thousands residents	85	13.4%
City with >100 thousands residents	246	38.9%

## Data Availability

The data presented in this study are available on request from corresponding author.

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
