# Peer review of "Colorectal Cancer—The Worst Enemy Is the One We Do Not Know"

_ijerph, 2023, doi:10.3390/ijerph20031866_

Round 1
Reviewer 1 Report
In the abstract, lines 13 and 14. " The article aimed at checking the knowledge of Polish 13 population on the above-mentioned topics". Please write the names of these topics; the study's aims must be clear for everyone.
In your introduction, it is better to write a short literature review ( previous studies that examine your study aim ( checking the knowledge of Polish 13 population on the ......). In addition, it is necessary to write the gap of knowledge of your studies ( what your studies will add to previous studies).
Conclusion
It is essential to write the limitation of your study.
Author Response
Dear Reviewer,
We are very grateful for the work you put into reviewing our article.
We have applied the given corrections (in abstract, introduction and conclusions) to our article and corrected syntax errors and past/present tenses errors.
Sincerely, Witold Zardzewiały

Reviewer 2 Report
thank you for allowing me to review this paper. the authors conducted a cohort survey on the population's knowledge of colorectal cancer. this study took place between 16 march and 1 may 2020.
only 631 questionnaires were usable; we would like to know the percentage of the population that responded, as the number is low. how many people were contacted to fill in the questionnaire? did the authors follow up with the people so that the percentage of responses was higher?
Did the authors calculate a minimum number of responses before sending out the questionnaire so that the sample would be representative?
What are the limitations of this study according to the authors? What prospects are envisaged to improve the knowledge of the population?
Author Response
Dear Reviewer,
We are very grateful for the work you put into reviewing our article.
We fixed typos with number of respondents.
At the lines 92-97 we clarified numbers of completed questionnaires.
Minimum numbers of responds also have been added at 97 line.
Limitations of our study have been added at line 374.
At lines 367-371 we proposed what could be done to further preventing detection of the disease at an advanced stage, thereby increasing the chance of a higher 5-year survival rate.
The revised manuscript is attached as an appendix.
Sincerely, Witold Zardzewiały

Reviewer 3 Report
The manuscript entitled “ Colon cancer - the worst enemy is the one we do not know” submitted by Niczyporuk et al. completed an online-based survey to evaluate the awareness of polish patients regarding colorectal cancer and authors divided the patients according to age, sex, education, employment and residence and compared them. The manuscript has some incomplete information and also flaws. My comments are as follows
1. The title does not represent the outcome of the manuscript. The percentage of rectum cancer is
52.1%, but the title represents only colon cancer. Besides, several subgroups have knowledge regarding colorectal cancer, including genetic predisposition- but the authors mentioned, “Colon cancer - the worst enemy is the one we do not know”
2. The authors mentioned the number of respondents is 631 in the abstract and Material- Methods sections. But according to table one, the respondents were 633.
3. The statistical information provided in the 2nd paragraph of the introduction was collected from Kuipers et al. (2015). Authors should present the latest data with proper citations.
4. Although in the introduction and screening sections authors mentioned the connection of CRC with family history, no data on family history was presented in the included cases.
5. The full questionnaire was absent in the main manuscript or as supplementary material.
6. Lines 123-125 create confusion as there is no common distribution of gender, age and education i.e., gender-wise age or education distribution was absent in the manuscript, although the authors claimed significant findings. Also, how the authors calculated ‘knowledge’ is not explicit.
7. Lines 128-129 also created confusion, where authors mentioned the knowledge of symptoms- but no data on symptoms were found in the text or table of the manuscript.
8. Lines 131-133 created confusion as no symptoms data was found in the manuscript either separately or employment-wise.
9. Lines 136-140: No data was found to support the occupational and gender-based knowledge data.
10. New results were presented in the different paragraphs of the discussion section (Eg. Lines 163-166, lines 176-177, 192, 212-214etc.) that were not presented in the result section or in the table. Moreover, the number of patients in line number 192 was mentioned as 636, which is also wrong.
11. No firm evidence was found either in the text data or tabular data in favor of the authors' claim in lines 241-251 of the conclusion section.
12. Some grammatical errors were found.
Author Response
Dear Reviewer,
We are very grateful for the work you put into reviewing our article.
-We renamed beggining of our article, as recommended.
-We fixed typos with number of respondents.
-As advised, the article from 2015 was replaced with three latest articles.
-Please check appendix in this message, questionnaire is added. The survey tested respondents' knowledge. The following three were used
questions - one per scale, assessing knowledge. For each correct answer one could get 2 points and an incorrect -1. We also attached the accurate statistical study at the appendix.
-earlier 128-133, now 198-215 lines we added data on colorectal cancer symptoms
-earlier 136-140, now 217-228 lines we added advised data
-we clarified typos error with number of respondents
I attached needed data at appendix in compressed document.
Sincerely, Witold Zardzewiały

Reviewer 4 Report
See attachment

Author Response
Dear Reviewer,
We are grateful for the work and effort put into improving our manuscript. We hope that our corrections will be satisfactory
-We have have corrected typos related to the number of respondents.
-We have added adjusted p-value at Bonferroni adjustment.
-We have revised the entire paragraph to include information about standardization by age and to correct an error about the most common cancer in the Western world
-With the division into primary, secondary and tertiary prevention we met, on the way of Polish education. If such a term does not exist in English-language nomenclature please forgive us. Primary prevention includes prevention through, for example, vaccination. Secondary prevention is screening. Tertiary prevention includes rehabilitation efforts to restore (or increase) function, help in coping with the limitations imposed by the disease, counteracting the patient's social isolation, further decline in mental and physical condition and relapse.
-We have added the missing part about the complication of colonoscopy as recommended.
-We have added information about number of participants in lines 92-93.
-We have changed formulation "higher" to "college level", as the other syntax errors
-We have added number of comparisons of Bonferroni adjustment in text.
-We have added information on what tests were used for a particular hypothesis at 115-116 lines.
-We have added information needed at 150-156 lines.
-Chi-square test had been used (lines 263-265)
Sincerely, Witold Zardzewiały

Round 2
Reviewer 3 Report
The revised manuscript entitled “Colorectal cancer - the worst enemy is the one we do not know” submitted by Furtak –Niczyporuk improved the quality tremendously from the previous version. The manuscript is now looking standard. I have the following queries regarding this manuscript.
1. There are some errors in the design the question no. 10,11,12,13,16. If the authors add ‘ I do not know’ we can get a concrete outcome from these questions. Due to the lack of this information, the authors were forced to choose other options for the multiple questions even though there was a chance the respondents did know the answers.
2. According to Figures 1,2,4,6, the total percentage of respondents crossed 100%, which might be due to the choice of multiple options. Although there was an option of choosing multiple answers in the question sections, the authors did not mention this in the manuscript text. If single respondents and multiple respondents can be identified (like internet, medical books, internet + Medical books etc.), we can get a finite idea regarding the outcomes.
3. Line 284-285 (blood in the stool will be 87.25% instead of 86.6%, and weight loss will be 61.45% instead of 61.2%, according to figure 4.
4. Still, some grammatical ( line 185: differing: correct corm: differ) and typing errors (knowsledge, section 3.1) were found.
Author Response
Dear Reviewer,
Thank you once again for your valuable guidance, which will help us in the future to avoid the mistakes made in this article. We especially thank you for your valuable comments on statistics and meticulousness. We have made the corrections that were recommended.
Sincerely, Witold Zardzewiały

Reviewer 4 Report
Overall, the authors have responded well to the issues raised in the initial review. However, A few additional changes would be beneficial.
1) Some additional figures are provided. While percentages are given for the responses, the sample sizes for the responses have not been given. If the number is 633 throughout the results section, that could be noted at the beginning of the section. Otherwise, the figure legend could indicate the sample size for each of the figures. Notably, all the rates in Figure 5 have a final digit of “0”, suggesting that N is not 633 for this survey item.
2) Figure 7, especially, could be reduced to text, if reduction of the article size is needed.
3) The author contributions section and other section at the end of the manuscript have not yet been completed.
Author Response
Dear Reviewer,
Thank you once again for such valuable comments that will help us not to make such mistakes in the future. We are grateful for your time and meticulous review of our article.
Sincerely, Witold Zardzewiały
